# Beyond Radiomics Alone: Enhancing Prostate Cancer Classification with ADC Ratio in a Multicenter Benchmarking Study

**DOI:** 10.3390/diagnostics15192546

**Published:** 2025-10-09

**Authors:** Dimitrios Samaras, Georgios Agrotis, Alexandros Vamvakas, Maria Vakalopoulou, Marianna Vlychou, Katerina Vassiou, Vasileios Tzortzis, Ioannis Tsougos

**Affiliations:** 1Medical Physics Department, Faculty of Medicine, University of Thessaly, 41500 Larissa, Greece; dimitsamaras@uth.gr; 2Archimedes Research Unit, Athena Research Center, 15125 Athens, Greece; maria.vakalopoulou@centralesupelec.fr; 3Department of Radiology, The Netherlands Cancer Institute, 1066 CX Amsterdam, The Netherlands; g.agrotis@uth.gr; 4Department of Anatomy and Neurosciences, Amsterdam UMC, Location Vrije Universiteit Amsterdam, 1081 HV Amsterdam, The Netherlands; alvamvakas@uth.gr; 5MICS Laboratory, CentraleSupélec, Université Paris-Saclay, 91190 Gif-sur-Yvette, France; 6Department of Radiology, Faculty of Medicine, University of Thessaly, 41500 Larissa, Greece; mvlychou@uth.gr; 7Department of Anatomy, Faculty of Medicine, University of Thessaly, 41500 Larissa, Greece; avassiou@uth.gr; 8Department of Urology, Faculty of Medicine, University of Thessaly, 41500 Larissa, Greece; vtzortzis@uth.gr

**Keywords:** prostate cancer, radiomics, machine learning, feature selection, ADC ratio, MRI, multicenter study, ComBat harmonization, classification

## Abstract

**Background/Objectives**: Radiomics enables extraction of quantitative imaging features to support non-invasive classification of prostate cancer (PCa). Accurate detection of clinically significant PCa (csPCa; Gleason score ≥ 3 + 4) is crucial for guiding treatment decisions. However, many studies explore limited feature selection, classifier, and harmonization combinations, and lack external validation. We aimed to systematically benchmark modeling pipelines and evaluate whether combining radiomics with the lesion-to-normal ADC ratio improves classification robustness and generalizability in multicenter datasets. **Methods**: Radiomic features were extracted from ADC maps using IBSI-compliant pipelines. Over 100 model configurations were tested, combining eight feature selection methods, fifteen classifiers, and two harmonization strategies across two scenarios: (1) repeated cross-validation on a multicenter dataset and (2) nested cross-validation with external testing on the PROSTATEx dataset. The ADC ratio was defined as the mean lesion ADC divided by contralateral normal tissue ADC, by placing two identical ROIs in each side, enabling patient-specific normalization. **Results**: In Scenario 1, the best model combined radiomics, ADC ratio, LASSO, and Naïve Bayes (AUC-PR = 0.844 ± 0.040). In Scenario 2, the top-performing configuration used Recursive Feature Elimination (RFE) and Boosted GLM (a generalized linear model trained with boosting), generalizing well to the external set (AUC-PR = 0.722; F1 = 0.741). ComBat harmonization improved calibration but not external discrimination. Frequently selected features were texture-based (GLCM, GLSZM) from wavelet- and LoG-filtered ADC maps. **Conclusions**: Integrating radiomics with the ADC ratio improves csPCa classification and enhances generalizability, supporting its potential role as a robust, clinically interpretable imaging biomarker in multicenter MRI studies.

## 1. Introduction

Prostate cancer (PCa) is the second most commonly diagnosed cancer in men and a major cause of cancer-related death worldwide [1]. Early detection significantly reduces mortality, with 5-year survival rates approaching 97% for localized disease [2]. Multiparametric MRI (mpMRI) has transformed PCa evaluation, improving lesion detection and localization, yet histopathological confirmation via biopsy remains the diagnostic gold standard [3]. Gleason scores (GS) derived from biopsy samples guide risk stratification but may differ from radical prostatectomy results, leading to under- or overgrading and contributing to over- and underdiagnosis [4,5].

Standard mpMRI protocols include T2-weighted imaging (T2w), diffusion-weighted imaging (DWI), and dynamic contrast-enhanced imaging (DCE) [6,7]. Among these, DWI-derived apparent diffusion coefficient (ADC) maps provide quantitative information on tumor cellularity and microstructure [8]. Lower ADC values generally indicate higher Gleason grades. The Prostate Imaging Reporting and Data System (PI-RADS) was developed to standardize interpretation, but distinguishing tumors from benign inflammatory lesions remains challenging [9,10].

A promising solution is the ADC ratio, defined as the mean ADC of the lesion divided by that of contralateral normal tissue [11]. This normalized metric accounts for interpatient and inter-scanner variability and better reflects tumor aggressiveness [12]. Lower ADC values indicate restricted diffusion from increased cellularity, whereas necrosis or atrophy may elevate ADC due to increased free water [8]. The ADC ratio accentuates these contrasts by normalizing lesion ADC to patient-specific background tissue, thereby emphasizing biologically relevant diffusion differences. Prior studies report that the ADC ratio can differentiate clinically significant (csPCa, GS ≥ 3 + 4) from insignificant disease (cinsPCa, GS ≤ 3 + 3) [13,14,15,16,17].

Radiomics has emerged as a complementary, AI-driven approach, extracting quantitative features from medical images to describe tumor phenotype beyond visual assessment [18,19,20,21,22]. The radiomics workflow includes image acquisition, optional preprocessing, ROI segmentation, feature extraction, and analysis using statistical or ML models [23]. By integrating radiomic features with interpretable biomarkers such as the ADC ratio, classification accuracy and generalizability may improve.

A persistent challenge is ensuring reproducibility across scanners and sites. Acquisition variability can confound radiomic feature distributions, motivating harmonization techniques such as ComBat to adjust for scanner effects and enhance comparability [24,25,26,27]. Dimensionality reduction is equally critical: filter, wrapper, and embedded feature selection (FS) methods [28,29] help retain informative features and prevent overfitting. Classifier choice also affects performance, with tree-based, linear, and kernel methods each offering strengths in different contexts [30,31,32].

Few large-scale studies systematically compare multiple FS methods and classifiers [33,34,35,36]. Recent work using the PROSTATEx dataset has evaluated reproducibility and external validity of radiomics workflows [37]. Complementary studies have examined the repeatability of prostate MRI radiomic features in the ProstateX cohort, highlighting the importance of selecting features that remain robust under segmentation or acquisition variability [38]. Together, these studies establish PROSTATEx as a reference benchmark for prostate MRI radiomics and support the need for systematic evaluation of FS methods, classifiers, and harmonization strategies, as performed in this study. Most rely on single pipelines and do not include simple, interpretable biomarkers like the ADC ratio. Incorporating such biomarkers may enhance model interpretability, especially in multicenter settings where scanner normalization is needed. Determining the optimal number of retained features is also crucial, as too few may discard valuable information, whereas too many risks overfitting; heuristics such as the “rule of 10” provide guidance [39]. Finally, ensemble approaches like voting classifiers may improve robustness by combining complementary models [33,40], although their advantage in multicenter radiomics remains unclear [41].

This study aims to develop robust ML models for csPCa classification by combining radiomics with the ADC ratio. We:Systematically evaluate nine different feature selection methods and multiple classifiers;Evaluate the additive value of the ADC ratio;Investigate the impact of ComBat harmonization and feature count;Perform repeated nested CV and external validation (PROSTATEx) to assess generalizability.

## 2. Materials and Methods

### 2.1. Patient Cohort

This study included two cohorts: the publicly available PROSTATEx dataset [42] and a local multicenter dataset collected from different centers. The inclusion of data acquired from different scanners with varying magnetic field strengths (1.5 T and 3.0 T) and acquisition protocols were intended to reflect real-world clinical heterogeneity and support the development of generalizable machine learning models for prostate cancer (PCa) diagnosis. The total cohort consisted of 209 patients with histologically confirmed PCa.

From the PROSTATEx dataset, 98 patients with PCa were included. These individuals underwent multiparametric MRI (mpMRI) using standardized protocols, with diffusion-weighted imaging (DWI) acquired at b-values of 50 and 800 s/mm^2^. Patients with PI-RADS scores ≥ 3 were referred for MRI-targeted biopsies. Histopathological evaluation was conducted by a uropathologist with over 20 years of experience [42]. In cases where more than one lesion was identified, all annotated lesions were considered, resulting in a total of 104 segmentations. Manual segmentations provided by previous studies were used (https://github.com/rcuocolo/PROSTATEx_masks (accessed on 5 October 2025)) [43,44].

The local dataset included 111 patients with biopsy-confirmed PCa who underwent mpMRI at collaborating centers. MpMRI was performed on 1.5 T or 3.0 T scanners using DWI protocols with b-values of 0 or 50 and 800 s/mm^2^. Patients were included if they had a PI-RADS score ≥ 3 based on the mpMRI report and had undergone both TRUS-guided systematic biopsy (12-core template) and MRI-targeted biopsy. Lesions were retrospectively identified and segmented on ADC maps by a board-certified radiologist with more than 20 years of experience using LIFEx (version 7.6.6) [45], incorporating T2w and DWI images and correlating with histopathology for anatomical precision. Segmentation was performed on the entire visible lesion, excluding surrounding benign tissue, peritumoral edema, and post-biopsy hemorrhage. This ensured that the ROI accurately captured the tumor signal without confounding effects from adjacent tissue changes.

MRI acquisition parameters for the local dataset were collected and summarized to ensure reproducibility. Table 1 provides the scanner vendor, model, field strength, diffusion b-values, repetition time (TR), echo time (TE), acquisition matrix, and slice thickness. Acquisition parameters for the PROSTATEx dataset followed the standardized mpMRI protocol reported in its documentation and in prior studies [42].

Exclusion criteria for both cohorts included poor image quality or incomplete lesion annotation. Seven patients were excluded from the local dataset due to poor DWI quality or lesions too small for diagnostic evaluation. An additional six patients were excluded from the PROSTATEx dataset due to annotation errors. The final dataset included 202 PCa lesions, comprising 54 clinically insignificant (cinsPCa, 26.7%) and 148 clinically significant (csPCa, 73.3%) lesions.

To calculate the ADC ratio, a normal reference region of interest (ROI) was placed contralaterally to each tumor ROI on the ADC map. Mean ADC values were computed for both the lesion and the normal ROI, and the ratio was defined as the mean ADC of the tumor divided by that of the reference region. All segmentations were conducted using LIFEx to ensure consistency and reproducibility across the dataset.

An example of lesion and contralateral ROI delineation on ADC maps is shown in Figure 1, illustrating the regions used to calculate the ADC ratio.

### 2.2. Feature Extraction and Pre-Processing

All preprocessing procedures were conducted using the Pyradiomics library (version 3.0.1) [46]. A detailed description of the preprocessing parameters is provided in Appendix A. For the purposes of this study, only apparent diffusion coefficient (ADC) maps derived from diffusion-weighted imaging (DWI) sequences with b-values of 0 or 50 and 800 s/mm^2^ were used, based on mono-exponential model fitting. These specific b-values were selected as they represented the common acquisition parameters across both the local and PROSTATEx datasets, ensuring consistency and comparability between cohorts.

To enhance spatial uniformity and feature reproducibility, all images were resampled to an isotropic voxel size of 1 mm^3^. Outlier voxel intensities were excluded prior to feature extraction using the μ ± 3σ criterion [47]. Intensity normalization was performed by scaling pixel values by a factor of 100. Fixed bin width discretization was set to 25 to optimize the reproducibility of texture features.

Radiomic feature extraction adhered to the guidelines of the Imaging Biomarkers Standardization Initiative (IBSI) [48]. Features extracted from each three-dimensional region of interest (3D ROI) included shape, first-order, and texture features based on gray-level co-occurrence matrix (GLCM), gray-level run length matrix (GLRLM), gray-level size zone matrix (GLSZM), gray-level dependence matrix (GLDM), and neighboring gray-tone difference matrix (NGTDM). In addition to features derived from the original ADC maps, higher-order features were obtained from images processed with Gradient filters, Laplacian of Gaussian (LoG) filters (with kernel sizes ranging from 2 mm to 5 mm), and Wavelet decompositions involving all combinations of high- and low-pass filtering across each spatial dimension.

Furthermore, the ADC ratio—a semi-quantitative imaging biomarker—was computed for each lesion. The ADC ratio was defined as the mean ADC value of the tumor ROI divided by the mean ADC value of a corresponding contralateral normal tissue ROI. In total, 1247 radiomic features were extracted per patient and per lesion.

To address inter-scanner and inter-protocol variability inherent to the multicenter nature of the dataset, ComBat harmonization was applied to the extracted features. Originally developed for genomics applications [49,50], ComBat employs an empirical Bayes framework to correct for batch effects while preserving biological signals. Harmonization was performed prior to any feature scaling to minimize scanner-related variability [51].

Following ComBat harmonization, z-normalization was performed to standardize the radiomic feature values. Each feature was scaled to have a mean of zero and a standard deviation of one based on the training data distribution. Z-score normalization was applied to ensure that features with different ranges and units contributed equally to the machine learning models, preventing bias toward features with larger numeric scales [52].

### 2.3. Feature Selection

Prior to feature selection, dimensionality reduction was performed to eliminate irrelevant and redundant features. Features with very low variance (threshold < 0.01) were removed, as they were unlikely to contribute meaningful discriminatory information. Additionally, highly correlated features were excluded based on Pearson’s correlation analysis; specifically, when the correlation coefficient between two features exceeded 0.85, the feature with the lower mean absolute correlation with the remaining feature set was retained. This step significantly reduced the dimensionality of the dataset, minimizing the risk of multicollinearity and enhancing the robustness of subsequent model training.

Following this initial filtering, the minimum redundancy maximum relevance (mRMR) algorithm was applied to further refine the feature set. mRMR was selected due to its ability to prioritize features that are highly correlated with the target outcome while minimizing inter-feature redundancy, thereby preserving the diversity and predictive potential of the selected variables. Based on this criterion, the 50 most informative features were retained for subsequent analysis.

Importantly, all supervised feature selection procedures were applied exclusively to the training data within each cross-validation fold to prevent data leakage and ensure unbiased model evaluation.

To identify the final subsets of predictive features, we evaluated a range of commonly used feature selection methods, as detailed in Appendix A. These methods were categorized into three major groups based on their selection strategy. Filter methods included the Analysis of Variance (ANOVA) and the Kruskal–Wallis test, which rank features according to their statistical association with the target outcome. Wrapper methods involved Recursive Feature Elimination (RFE), Backward Feature Selection (BFS), Sequential Forward Selection (SFS), and Boruta, which assess feature subsets based on model performance. Finally, embedded methods consisted of L1-regularized Least Absolute Shrinkage and Selection Operator (LASSO) and Random Forest-based variable importance (RF-imp), which integrate feature selection within the model training process. Specifically, ANOVA, BFS, SFS, Kruskal–Wallis, RFE, and RF-imp were used to select varying numbers of top-ranked features (5, 10, 15, and 20 features), in order to investigate the impact of feature count on model performance. This design was informed by the “rule of 10,” which recommends maintaining at least 10 samples per feature to reduce the risk of overfitting, particularly in high-dimensional datasets.

In contrast to methods with predefined feature subset sizes, Boruta and LASSO are data-driven embedded approaches that determine the optimal number of features through internal relevance metrics or regularization strength. Specifically, Boruta iteratively compares the importance of real features against shadow (randomized) counterparts to identify all-relevant features, while LASSO imposes an L1 penalty to shrink irrelevant feature coefficients to zero. As a result, each method yielded a single, algorithmically determined feature subset without manual tuning of feature count parameters.

The applied feature selection techniques represent complementary approaches: filter-based methods (e.g., ANOVA, Kruskal–Wallis) rank features by univariate association, wrapper-based methods (e.g., RFE, SFS, BFS) iteratively evaluate subsets using model performance, and embedded methods (e.g., LASSO, RF-imp) select features during model training. A detailed overview of these methods is provided in Appendix A.

### 2.4. Classification Modelling and Assessment

Following feature selection, classification models were developed to evaluate the diagnostic performance of the selected radiomic features and ADC ratio for distinguishing clinically significant prostate cancer (csPCa) from clinically insignificant prostate cancer (cinsPCa).

Three distinct types of models were trained for each feature selection method:A radiomics model, based solely on radiomic features extracted from ADC maps.A radiomics–ADC ratio model, combining the selected radiomic features with the ADC ratio.An ADC ratio stand-alone model, using only the ADC ratio as the input feature.

All machine learning algorithms were implemented using the Scikit-learn library (version 1.3.2) in Python. For each feature selection method, multiple classifiers were trained, including linear models, tree-based models, support vector machines (SVM), and ensemble methods. A complete list of the classifiers, along with their respective configurations and hyperparameters, is provided in Appendix A.

Hyperparameter optimization for each classifier was performed using a grid search strategy embedded within the cross-validation loop to avoid information leakage. Similarly, all supervised feature selection steps were performed exclusively within the training data of each inner cross-validation fold to prevent information leakage into model evaluation. To control model complexity and reduce overfitting risk, a repeated nested cross-validation (CV) design was employed. The inner CV loop was used for hyperparameter tuning and feature selection, while the outer CV loop provided an unbiased estimate of model performance, ensuring strict separation of training and validation data. This approach is widely recommended for high-dimensional radiomics studies as it minimizes optimistic bias and prevents feature-selection leakage.

Two distinct validation scenarios were implemented to ensure robust model assessment:In Scenario 1, repeated stratified nested cross-validation (5 folds × 3 repeats for the outer loop and 5 folds × 10 repeats for the inner loop) was applied to the merged multicenter dataset.In Scenario 2, repeated stratified nested cross-validation (4 folds × 3 repeats for the outer loop and 5 folds × 1 repeats for the inner loop) was performed exclusively on the local dataset, followed by external testing on the PROSTATEx dataset to investigate generalization of the model.

Classification models were evaluated using multiple metrics, including the area under the precision–recall curve (AUC-PR), F1-score, area under the receiver operating characteristic curve (AUC-ROC), precision, and recall. AUC-PR was selected as the primary metric given its suitability for imbalanced data, with AUC-ROC reported as a complementary measure of discriminative ability. For feature selection methods that allowed manual control over the number of selected features (ANOVA, BFS, SFS, Kruskal–Wallis, RFE, and RF-imp), models were trained and evaluated using 5, 10, 15, and 20 features to assess the impact of feature subset size on performance. For Boruta and LASSO, where the number of features was determined automatically, a single feature set was used.

Additionally, to evaluate whether ensemble learning could improve predictive robustness, voting classifiers were constructed by combining the three best-performing individual classifiers for each feature selection method. Both soft voting (based on predicted probabilities) and hard voting (based on predicted class labels) were explored.

### 2.5. Statistical Analysis

Descriptive statistics were reported as mean values with standard deviations. Model performance was primarily evaluated using the area under the precision–recall curve (AUC-PR), a metric particularly suitable for imbalanced classification problems such as the present study. Performance comparisons were conducted across validation results to identify the best-performing model configurations. To compare the performance of best-performing models across training scenarios, the Friedman test was used to assess statistically significant differences in AUC-PR values across models evaluated on the same cross-validation folds. The Friedman test is the non-parametric analogue of repeated-measures ANOVA and is recommended for comparing multiple models across cross-validation folds when normality cannot be assumed. When the Friedman test indicated a global difference, post hoc pairwise comparisons were performed using the Nemenyi test. Additionally, Wilcoxon signed-rank tests were applied for focused pairwise comparisons without assuming normality. All tests were two-sided, with *p*-values < 0.05 considered statistically significant.

To evaluate model generalization on the unseen test set (Scenario 2), permutation testing was employed. In this approach, class labels were randomly shuffled, and the performance metric of interest (e.g., AUC-PR) was recalculated for each permutation. The distribution of these permuted statistics was then compared to the observed statistics to determine whether the model’s performance was statistically significant. Statistical significance was defined as a two-tailed *p*-value less than 0.05.

This study adhered to established guidelines aimed at enhancing the credibility, reproducibility, and transparency of radiomics research. In particular, the Checklist for EvaluAtion of Radiomics Research (CLEAR) [53], the METhodological RadiomICs Score (METRICS) [54] and the Transparent Reporting of a multivariable prediction model for Individual Prognosis Or Diagnosis (TRIPOD) [55,56] were employed to systematically assess study quality. The study achieved a METRICS quality score classified as “Good” (71.91%). Specific results from the CLEAR checklist evaluation and the detailed METRICS scoring are provided in Appendix A.

The whole workflow is presented in Figure 2.

## 3. Results

### 3.1. Best Performing Model Across Scenarios

In Scenario 1, which employed repeated CV on the merged dataset, the best-performing configuration was the Radiomics–ADC ratio model without ComBat harmonization, using LASSO for feature selection and Naïve Bayes as the classifier. This combination achieved the highest AUC-PR (0.844 ± 0.040) and a notably improved F1-score (0.725 ± 0.079) compared to all other configurations (Table 2). In contrast, the standalone ADC ratio model resulted in a lower AUC-PR (0.754 ± 0.041) and F1-score (0.574 ± 0.116) (Table 3), confirming the benefit of combining radiomic features with the ADC ratio.

In Scenario 2, which training on the local dataset and external testing on the PROSTATEx dataset, the best performance was achieved by the Radiomics–ADC ratio model without ComBat, using RFE with 20 features selected and a Boosted GLM classifier. It yielded an internal AUC-PR of 0.885 ± 0.042 and F1 = 0.824 ± 0.043 and generalized best to the external test set (AUC-PR = 0.722, F1 = 0.741) (Table 4). Among all tested models, this configuration provided the most reliable balance between training performance and generalization. The standalone ADC ratio model also performed well in terms of F1 (0.802) but showed a lower AUC-PR (0.668), suggesting that it cannot fully substitute for radiomics-based modeling (Table 5). A detailed performance assessment for each classifier and feature selection method is provided in Appendix A.

The tuned hyperparameters for the best-performing models in Scenarios 1 and 2 are summarized in Appendix A, respectively. These parameters, optimized using grid search in the inner cross-validation loop based on mean AUC-PR, were used for final training and evaluation.

### 3.2. Impact of ComBat Harmonization

ComBat harmonization was evaluated as a mechanism for reducing inter-scanner variability. In Scenario 1, although models using ComBat (Radiomics_ComBat + BFS_20 + Random Forest, AUC-PR = 0.837 ± 0.061 and Radiomics-ADC ratio_ComBat + KW_20 + Ada Boost, AUC-PR = 0.839 ± 0.057) did not achieve the highest AUC-PR, they yielded better F1-scores than their unharmonized counterparts, suggesting improved balance between sensitivity and precision (Table 2). In Scenario 2, ComBat-enhanced models also performed competitively. For instance, the Radiomics_ComBat model using Extra Trees achieved an internal F1 of 0.864 ± 0.022, higher than most non-harmonized models, and retained good generalization (AUC-PR = 0.714; F1 = 0.811) (Table 4).

Overall, ComBat harmonization modestly improved model calibration (F1) but not peak AUC-PR, especially for models integrating the ADC ratio.

### 3.3. Generalization on External Data

Scenario 2 enabled external testing on an independent cohort (PROSTATEx). The Radiomics–ADC ratio model without ComBat, using RFE with the top 20 selected and Boosted GLM, yielded the highest external performance, achieving an AUC-PR of 0.722 and an F1-score of 0.741. In comparison, ComBat-harmonized models showed slightly lower external AUC-PRs—0.714 for the Radiomics + ComBat model and 0.676 for the Radiomics–ADC ratio + ComBat model—but retained relatively high F1-scores (0.811 and 0.765, respectively), suggesting improved class balance.

These findings indicate that while ComBat may improve internal prediction stability, its benefit for external AUC-PR is limited. The inclusion of the ADC ratio appears particularly beneficial for cross-site generalization when combined with well-selected radiomics features (Table 4).

### 3.4. Voting Classifiers vs. Single Classifiers

Voting ensemble classifiers were compared against the top-performing single-model counterparts across all scenarios (Figure 3a–d). In all configurations, voting classifiers failed to outperform single classifiers in terms of AUC-PR. While ensemble methods may offer theoretical advantages in terms of decision stability, their application in this study did not yield improvements in discriminative performance. In some cases, voting even resulted in marginal decreases in AUC-PR, particularly when combined with ComBat-harmonized features. These findings suggest that, for this classification task, ensemble voting offers limited utility over well-optimized single classifiers.

### 3.5. Classifier Comparison

Classifier performance was primarily evaluated using AUC-PR, with F1-score used as a complementary metric to assess class balance. As shown in Table 6 and Table 7, Random Forest, Boosted GLM, and Naïve Bayes emerged as the most robust and stable classifiers across both scenarios.

In Scenario 1, Naïve Bayes achieved the highest AUC-PR for both Radiomics (0.820 ± 0.010) and Radiomics–ADC ratio (0.824 ± 0.011) models without ComBat. Although its F1-scores were lower (~0.64), its simplicity and reproducibility make it a reliable baseline, especially in lower-dimensional settings.

Random Forest demonstrated superior F1-scores, particularly in ComBat-harmonized models (F1 = 0.801 ± 0.015 for Radiomics and 0.806 ± 0.021 for Radiomics–ADC ratio), while maintaining competitive AUC-PR values. This suggests its strength in producing more balanced predictions under scanner harmonization.

In Scenario 2, Random Forest again achieved the highest AUC-PRs, particularly for the Radiomics–ADC ratio model without ComBat (0.860 ± 0.013) and the Radiomics model (0.852 ± 0.017), both with consistent F1-scores (0.840 ± 0.008). Boosted GLM, used for ComBat models, showed slightly lower AUC-PRs but very stable performance across configurations (e.g., Radiomics–ADC ratio model: AUC-PR = 0.840 ± 0.016, F1 = 0.835 ± 0.016).

### 3.6. Feature Selection Methods

Multiple feature selection strategies were applied to reduce dimensionality and retain the most informative features for model training. These included filter-based (e.g., ANOVA, Kruskal–Wallis), wrapper-based (e.g., RFE, BFS, SFS), and embedded methods (e.g., LASSO, Random Forest importance). Their comparative performance was assessed based on AUC-PR and F1-score across both scenarios.

In both Scenario 1 and Scenario 2, Recursive Feature Elimination (RFE) emerged as one of the most effective and consistent methods, particularly when combined with ensemble classifiers such as Random Forest or Boosted GLM. For example, the highest-performing model overall (Radiomics–ADC ratio without ComBat in Scenario 2) used RFE_20, achieving an AUC-PR of 0.885 ± 0.042 and F1 = 0.824 ± 0.043.

Other top-performing methods included LASSO, which selected sparse feature sets and led to strong results in Scenario 1 (AUC-PR = 0.844 ± 0.040), and Kruskal–Wallis (KW_20), which performed well in ComBat-harmonized models. In contrast, BFS and SFS demonstrated more variable results, particularly with smaller feature counts, where performance tended to drop—highlighting the importance of choosing appropriate subset sizes.

Overall, RFE, LASSO, and Kruskal–Wallis offered the best trade-off between model accuracy and stability across different harmonization schemes and classification pipelines. Their consistent selection of robust radiomic descriptors across folds and scenarios suggests their strong suitability for radiomics-based classification of clinically significant prostate cancer.

### 3.7. Influence of Feature Count

The number of selected features played an important role in model performance. The best-performing configurations in both scenarios (Table 2 and Table 4) consistently used feature sets with approximately 20 variables, particularly when using RFE_20, ANOVA_20, KW_20, or BFS_20. For example, the top-performing Radiomics–ADC ratio model in Scenario 2 used RFE_20, while other high-performing models also relied on ANOVA_20 (Radiomics model) and KW_20 (Radiomics–ADC ratio + ComBat). In contrast, models using smaller subsets—such as BFS_5—achieved lower external AUC-PR (e.g., 0.676 vs. 0.722 with RFE_20), despite strong internal performance.

These findings are consistent with the “rule of 10,” which recommends a minimum of 10 samples per feature to ensure generalizability and reduce overfitting, particularly in small datasets. Feature sets with very few variables often failed to capture the complexity of tumor heterogeneity and underperformed in both internal and external validation, reinforcing the need for sufficient descriptive capacity in radiomics-based models.

### 3.8. Stable and Frequently Selected Features

Analysis of the most frequently selected features revealed consistent patterns, with the top 10 predominantly texture-based (GLCM, GLSZM), especially from Wavelet and LoG filters. Shape (Original_shape_Sphericity) and first-order (Wavelet_LLL_firstorder_TotalEnergy) features were also among the most robust across harmonization strategies and validation scenarios. Because feature subsets vary across the many combinations tested (multiple feature selection methods, classifiers, harmonization schemes, and repeated cross-validation folds), we report the ten most frequently selected features as the most stable radiomic biomarkers (Table 8).

### 3.9. Combined Model Performance

A combined model trained using the top 10 most frequently selected features was also evaluated. In Scenario 1, it achieved AUC-PR = 0.860 ± 0.045 and F1 = 0.798 ± 0.044 (Table 9), while in Scenario 2, it reached AUC-PR = 0.921 ± 0.059 on internal validation and 0.697 on external testing, with corresponding F1 = 0.868 ± 0.029 (train) and 0.816 (test) (Table 10). These results confirm the discriminative power and stability of these features across settings.

### 3.10. Statistical Comparison of Model Performance

Statistical comparisons using Friedman, Wilcoxon, and Nemenyi tests were performed on the best-performing models (Table 11 and Table 12, Figure 4, Figure 5 and Figure 6). In Scenario 1, no significant differences were observed (Friedman *p* = 0.6325). In Scenario 2, Friedman testing revealed significant differences (*p* = 0.0047), and post hoc Nemenyi analysis showed that the combined model significantly outperformed both the Radiomics_No_ComBat and Radiomics_ComBat models (*p* = 0.0167 and *p* = 0.0045, respectively). These results support the superiority of the combined feature approach in multicenter external testing.

In addition, permutation testing was performed on the external test set to assess whether differences in AUC-PR between models could have occurred by chance (Table 13). Despite the combined model yielding the highest AUC-PR on the external set, none of the pairwise comparisons reached statistical significance at *p* < 0.05.

Together, these statistical findings reinforce the robustness of the combined model and support the conclusion that its performance advantages are consistent, though not universally significant in external testing.

## 4. Discussion

This study presents a comprehensive and systematic evaluation of radiomics-based machine learning models for prostate cancer classification, benchmarking a diverse array of feature selection methods, classifiers, and harmonization strategies across both internal and external validation settings. By assessing over one hundred unique model configurations and conducting external validation using the PROSTATEx dataset, our study offers a uniquely broad and robust comparison, going beyond the scope of most prior studies that typically evaluate only a limited set of pipeline options [51,57,58].

A key innovation of this work lies in the integration of the ADC ratio—a simple, interpretable, and normalized imaging biomarker—into radiomics classification pipelines. Defined as the ratio of lesion ADC to contralateral healthy tissue, this feature provides biologically meaningful contrast while mitigating patient- and scanner-specific variability [59]. Unlike statistical harmonization approaches such as ComBat, the ADC ratio does not rely on site-specific correction and can be computed directly from the image, making it both reproducible and clinically intuitive. Notably, models incorporating the ADC ratio consistently outperformed radiomics-only or ADC-only models in both validation scenarios.

The best performing configuration in Scenario 1 was the Radiomics–ADC ratio model without ComBat, using LASSO feature selection and Naïve Bayes, which achieved an AUC-PR of 0.844 ± 0.040 and an F1-score of 0.725 ± 0.079. In Scenario 2, the best generalization was achieved by a model using RFE_20 + Boosted GLM, yielding a training AUC-PR of 0.885 and testing AUC-PR of 0.722. This confirms that combining handcrafted radiomic features with domain-normalized biomarkers like the ADC ratio enhances both discriminative power and external robustness.

It is important to interpret these F1-scores in the context of the underlying class distribution (73.3% csPCa vs. 26.7% cinsPCa). Although absolute values may appear high due to class imbalance, they provide a valid basis for relative model comparisons under identical stratified cross-validation splits. To mitigate prevalence effects, we emphasized AUC-PR as the primary metric, which better reflects the trade-off between precision and recall for the clinically relevant positive class.

While ComBat harmonization improved model calibration (e.g., higher F1-scores), it did not consistently enhance external AUC-PR and occasionally reduced discriminative performance. This finding is noteworthy because ComBat is one of the most widely used statistical harmonization techniques in radiomics and is often assumed to improve cross-site generalization [48]. The most likely explanation is that ComBat reduces inter-scanner variability by aligning the means and variances of radiomic features across scanners, but this adjustment does not necessarily change the relative ordering of patient-level predictions, which is what drives AUC-PR. As a result, the model’s ability to rank csPCa versus cinsPCa cases remains similar, leaving discrimination unchanged. However, by stabilizing the feature distributions and reducing site-specific bias, ComBat makes the predicted probabilities more consistent across scanners, thereby improving probability calibration and threshold-dependent metrics such as F1-score. In some cases, this linear correction may even suppress weak biological signals if scanner effects and disease effects are partially collinear, leading to slight reductions in discriminative power on external data. This highlights that linear harmonization may be insufficient to address nonlinear scanner effects—particularly in external validation scenarios [60]. In contrast, the ADC ratio provided a form of intrinsic normalization that generalized more effectively across scanners, indicating its potential as a surrogate for more complex harmonization techniques.

Among classifiers, Random Forest, Boosted GLM, and Naïve Bayes demonstrated consistent and high performance aligning with previous findings [34,51,58,61]. Random Forest performed particularly well in radiomics-only models with harmonization, while Boosted GLM excelled when combined with RFE and ADC ratio in external testing. Naïve Bayes, despite its simplicity, remained competitive and achieved top results in Scenario 1, validating its utility in low-dimensional radiomics applications.

Regarding feature selection, the most effective methods included RFE, LASSO, and Kruskal–Wallis, each contributing to top-performing models [34,61,62,63]. These techniques consistently selected informative features while avoiding redundancy. Feature count also played a critical role: models using ~20 features (e.g., RFE_20, ANOVA_20, KW_20) consistently outperformed those with fewer (e.g., BFS_5), particularly in external testing. These results align with the “rule of 10” guideline, which emphasize maintaining an adequate sample-to-feature ratio to minimize overfitting [18,39].

To explore model simplification, we trained a combined model using the top 10 most frequently selected features across all scenarios. This reduced model achieved competitive results—AUC-PR = 0.860 (Scenario 1) and 0.697 (Scenario 2 external), with F1-scores as high as 0.816. These findings demonstrate that compact, biologically plausible feature sets can maintain high performance, potentially increasing model interpretability and ease of clinical deployment.

Although we explored ensemble learning through hard and soft voting classifiers, these ensembles did not consistently outperform the best single classifier across feature selection methods or validation scenarios. This outcome likely reflects the fact that the top-performing individual models already captured most of the available discriminative signal, leaving limited opportunity for performance gains through aggregation. Prior radiomics studies have reported similar findings, suggesting that ensemble benefits are modest when base models are highly correlated or when one classifier is near optimal [40]. Carefully tuned single classifiers, when paired with appropriate feature selection, remain more effective for this task.

In addition to performance trends, statistical comparisons provided further validation. In Scenario 2, Friedman testing revealed significant differences among models (*p* = 0.0047), and Nemenyi post hoc tests confirmed that the combined model significantly outperformed radiomics models with and without ComBat harmonization. Although permutation tests on the external set showed no significant pairwise differences—likely due to limited test set size—the consistent ranking of the combined model reinforces its potential utility.

This study also reinforces the relevance of specific radiomic features. The most frequently selected features (Table 8) were largely texture-based (GLCM, GLSZM) and derived from Wavelet or LoG filters, alongside a few first-order and shape features (e.g., Total Energy, Sphericity) [34,36]. GLCM features such as Inverse Difference Moment Normalized (IDMN) and Inverse Difference Normalized (IDN) quantify local homogeneity by weighting gray-level differences inversely and normalizing for the number of gray levels. Higher values of IDMN/IDN correspond to greater gray-level uniformity, which may reflect more organized glandular structures, whereas lower values indicate increased heterogeneity—a hallmark of aggressive prostate cancer. GLSZM features such as Size-Zone Non-Uniformity Normalized (SZNN) describe the variability in homogeneous zone sizes, with higher values indicating more heterogeneous tissue architecture. Together, these features capture microstructural disorganization and cellular heterogeneity that correlate with tumor aggressiveness and higher Gleason patterns, supporting their biological plausibility as imaging biomarkers. Despite its strengths, this study has limitations. First, the external validation cohort was modest in size, limiting statistical power in detecting significant differences between close-performing models. Future research should involve larger, more diverse, and multi-institutional datasets. Second, the assumption of linear, additive batch effects underlying ComBat harmonization may be overly simplistic for complex imaging data. Future research should investigate more advanced harmonization strategies, including nonlinear or deep learning-based approaches, which may better capture the intricate relationships between scanner variability and radiomic features.

Third, while this study focused on features derived from ADC maps, incorporating additional modalities (e.g., T2w, DCE, PET) and clinical data (e.g., PSA, PI-RADS) could further improve robustness. Lastly, integrating radiomics with emerging AI frameworks like Visual Language Models (VLMs) may enable multimodal reasoning, enhancing generalization and interpretability [64].

In conclusion, this work demonstrates that prostate cancer radiomics classification can be substantially improved by combining handcrafted features with normalized, interpretable biomarkers like the ADC ratio. The proposed framework achieved strong performance across diverse configurations and generalized well to external data, setting a foundation for clinically deployable AI models in oncologic imaging.

## 5. Conclusions

This study systematically evaluated radiomics-based models and ADC ratio for classifying clinically significant prostate cancer using mpMRI. Incorporating the ADC ratio consistently improved model performance and robustness across scenarios. Comparative analysis of eight feature selection methods and fifteen classifiers showed that the choice of algorithm significantly influences results, with RFE, LASSO, and Kruskal–Wallis offering the best trade-offs between performance and stability. The model using the most frequently selected features across different feature selection methods achieved the highest AUC-PR, supporting the integration of interpretable biomarkers and optimized pipelines for clinically applicable radiomics models.

## Figures and Tables

**Figure 1 diagnostics-15-02546-f001:**
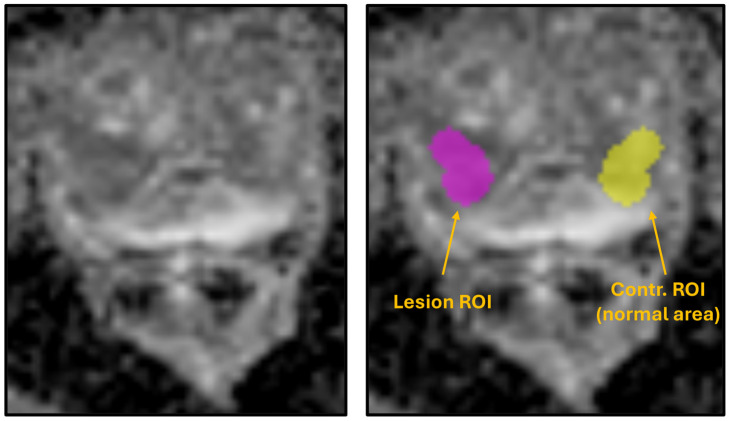
Example ADC map showing lesion ROI (purple) and contralateral normal tissue ROI (yellow) used for ADC ratio calculation.

**Figure 2 diagnostics-15-02546-f002:**
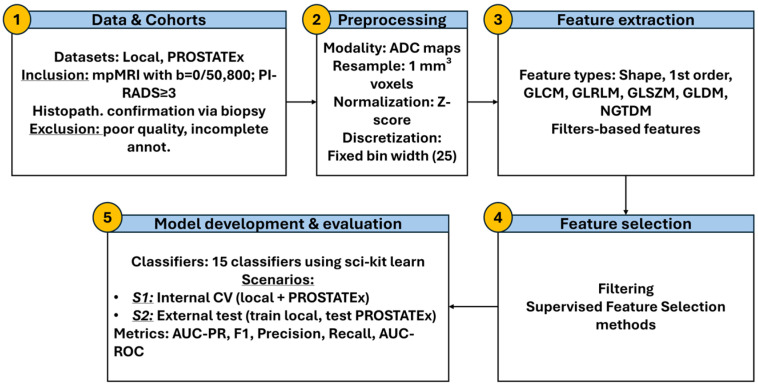
Overview of the radiomics pipeline used in this study, comprising five steps: (1) data acquisition from local and PROSTATEx cohorts; (2) ADC preprocessing; (3) IBSI-compliant radiomic and ADC ratio feature extraction; (4) feature selection via filtering and machine learning methods; and (5) model training and evaluation using internal and external validation.

**Figure 3 diagnostics-15-02546-f003:**
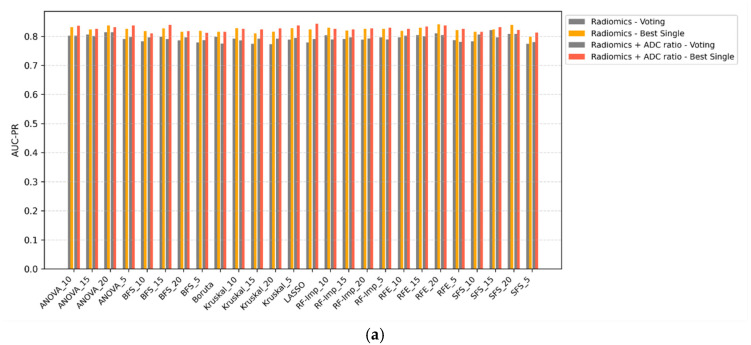
(**a**) Comparison of AUC-PR scores between voting classifiers and best single classifiers for Radiomics and Radiomics + ADC ratio models in Scenario 1 without ComBat harmonization. Voting classifier bars are colored green when outperforming the corresponding single model. (**b**) Comparison of AUC-PR scores between voting classifiers and best single classifiers for Radiomics and Radiomics + ADC ratio models in Scenario 1 after ComBat harmonization. Voting classifier bars are colored green when outperforming the corresponding single model. (**c**) Comparison of AUC-PR scores between voting classifiers and best single classifiers for Radiomics and Radiomics + ADC ratio models in Scenario 2 without ComBat harmonization. Voting classifier bars are colored green when outperforming the corresponding single model. (**d**) Comparison of AUC-PR scores between voting classifiers and best single classifiers for Radiomics and Radiomics + ADC ratio models in Scenario 2 after ComBat harmonization. Voting classifier bars are colored green when outperforming the corresponding single model.

**Figure 4 diagnostics-15-02546-f004:**
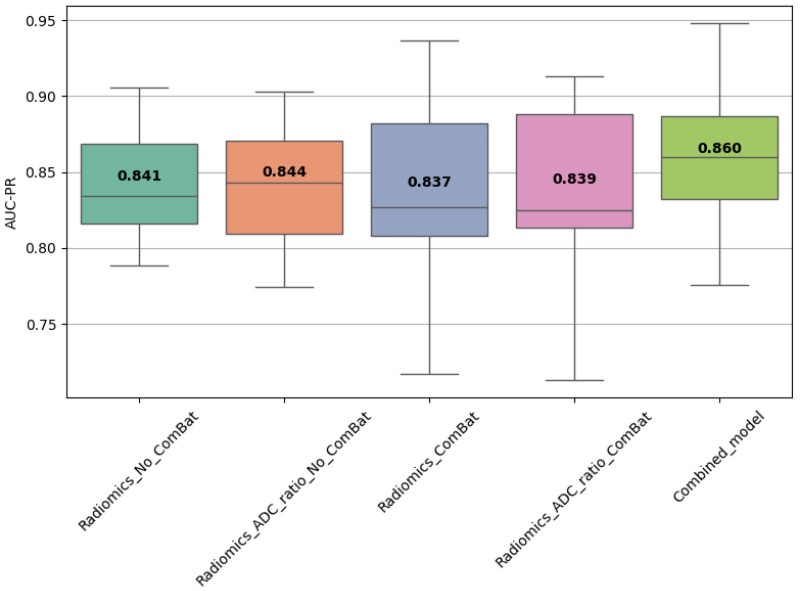
Distribution of AUC-PR values across the best-performing models in each training scenario (Scenario 1). Each boxplot represents the AUC-PR values obtained from CV folds. Bold numbers within each box indicate the mean AUC-PR for the corresponding model.

**Figure 5 diagnostics-15-02546-f005:**
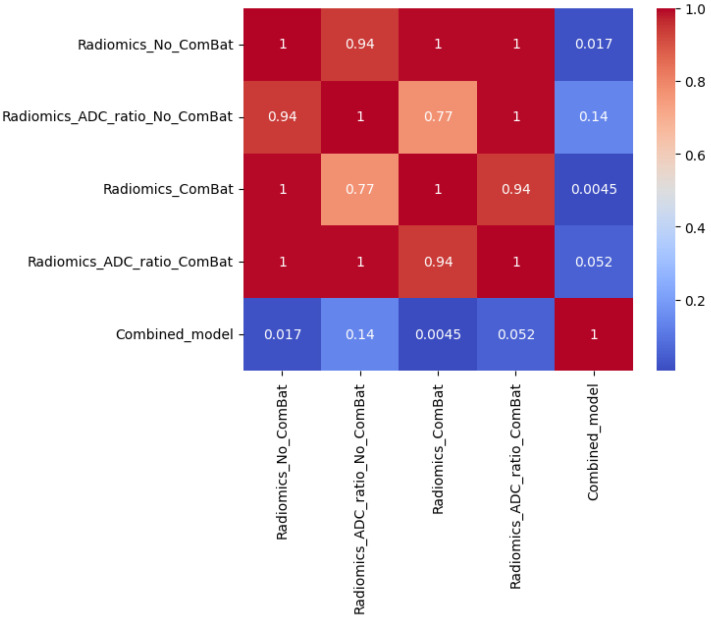
Heatmap of pairwise *p*-values from the Nemenyi post hoc test comparing AUC-PR distributions across five models in Scenario 2. Each cell shows the *p*-value for the comparison between two models; darker blue values indicate stronger evidence of difference. The Combined_model significantly outperformed both Radiomics_No_ComBat (*p* = 0.017) and Radiomics_ComBat (*p* = 0.0045), while no other pairwise comparisons were statistically significant (*p* ≥ 0.05).

**Figure 6 diagnostics-15-02546-f006:**
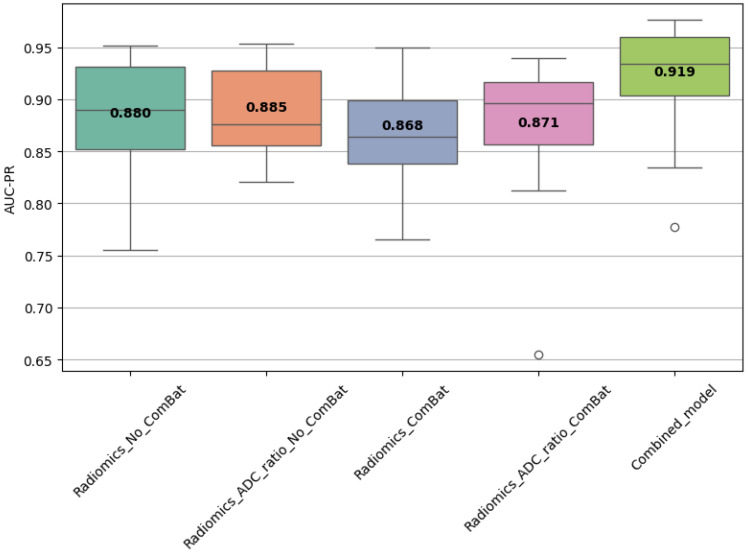
AUC-PR distribution across best-performing models in Scenario 2. Boxes represent values from cross-validation folds; bold numbers indicate mean AUC-PR. A significant difference was found (Friedman test, *p* = 0.0047), with post hoc tests showing the combined model outperformed the other models.

**Table 1 diagnostics-15-02546-t001:** MRI acquisition parameters for scanners used in the local dataset.

Scanner	Patients	Vendor	Model	Field Strength	b-Values, s/mm^2^	TR, ms	TE, ms	Acq. Matrix	Slice Thick., mm
1	38	GE (Boston, MA, USA)	Signa HDxt	3.0 T	0, 800	5050	71.6	256 × 256	3
2	27	Siemens (Munich, Germany)	Magnetom Vida	3.0 T	50, 800	6360	58	236 × 236	3
3	33	Siemens	Avanto	1.5 T	0, 800	4000	60	112 × 112	3.5

**Table 2 diagnostics-15-02546-t002:** Scenario 1—Best Model Performance.

Scenario 1						
Model	Harmonization Scheme	Feature Selection	Classifier	AUC-PR	F1	AUC-ROC
Radiomics model	No ComBat	RFE_20	Naïve Bayes	0.841 ± 0.033	0.574 ± 0.116	0.633 ± 0.067
Radiomics-ADC ratio model	No ComBat	LASSO	Naïve Bayes	0.844 ± 0.040	0.725 ± 0.079	0.641 ± 0.063
Radiomics model	ComBat	BFS_20	Random Forest	0.837 ± 0.061	0.813 ± 0.039	0.619 ± 0.119
Radiomics-ADC ratio model	ComBat	KW_20	Ada Boost	0.839 ± 0.057	0.777 ± 0.051	0.626 ± 0.102

**Table 3 diagnostics-15-02546-t003:** Scenario 1—ADC ratio Model Performance.

Scenario 1			
Model	AUC-PR	F1	AUC-ROC
ADC ratio model	0.754 ± 0.041	0.574 ± 0.116	0.519 ± 0.073

**Table 4 diagnostics-15-02546-t004:** Scenario 2—Best Model Performance.

		Scenario 2							
		Training Set					Test Set		
Model	Harmonization Scheme	Feature Selection	Classifier	AUC-PR	F1	AUC-ROC	AUC-PR	F1	AUC-ROC
Radiomics model	No ComBat	ANOVA_20	Random Forest	0.880 ± 0.061	0.837 ± 0.051	0.653 ± 0.107	0.717	0.748	0.521
Radiomics-ADC ratio model	No ComBat	RFE_20	Boosted GLM	0.885 ± 0.042	0.824 ± 0.043	0.651 ± 0.098	0.722	0.741	0.512
Radiomics model	ComBat	SFS_20	Extra Trees	0.868 ± 0.054	0.864 ± 0.022	0.638 ± 0.087	0.714	0.811	0.501
Radiomics-ADC ratio model	ComBat	BFS_5	Boosted GLM	0.871 ± 0.079	0.832 ± 0.030	0.640 ± 0.145	0.676	0.765	0.473

**Table 5 diagnostics-15-02546-t005:** Scenario 2—ADC ratio Model Performance.

	Scenario 2					
Training Set				Test Set		
Model	AUC-PR	F1	AUC-ROC	AUC-PR	F1	AUC-ROC
ADC ratio model	0.808 ± 0.078	0.847 ± 0.024	0.568 ± 0.156	0.668	0.802	0.426

**Table 6 diagnostics-15-02546-t006:** Scenario 1—Comparison Across Classifiers.

Model	Harmonization Scheme	Classifier	AUC-PR	F1	AUC-ROC
Radiomics model	No ComBat	Naïve Bayes	0.820 ± 0.010	0.646 ± 0.044	0.611 ± 0.015
Radiomics-ADC ratio model	No ComBat	Naïve Bayes	0.824 ± 0.011	0.641 ± 0.044	0.611 ± 0.021
Radiomics model	ComBat	Random Forest	0.813 ± 0.014	0.801 ± 0.015	0.583 ± 0.022
Radiomics-ADC ratio model	ComBat	Random Forest	0.807 ± 0.016	0.806 ± 0.021	0.573 ± 0.024

**Table 7 diagnostics-15-02546-t007:** Scenario 2—Comparison Across Classifiers.

Model	Harmonization Scheme	Classifier	AUC-PR	F1	AUC-ROC
Radiomics model	No ComBat	Random Forest	0.852 ± 0.017	0.840 ± 0.008	0.607 ± 0.037
Radiomics-ADC ratio model	No ComBat	Random Forest	0.860 ± 0.013	0.840 ± 0.008	0.618 ± 0.028
Radiomics model	ComBat	Boosted GLM	0.841 ± 0.010	0.834 ± 0.015	0.580 ± 0.019
Radiomics-ADC ratio model	ComBat	Boosted GLM	0.840 ± 0.016	0.835 ± 0.016	0.580 ± 0.033

**Table 8 diagnostics-15-02546-t008:** Top 10 most frequently selected radiomic features across all models and scenarios. Features follow IBSI-compliant nomenclature and span first-order, shape, and texture families. Texture features from Wavelet and LoG filters were most dominant, reflecting their relevance for characterizing tumor heterogeneity in csPCa.

Feature Name	Filter Type	Category
Original_shape_Sphericity	Original	Shape
Wavelet_LLL_firstorder_TotalEnergy	Wavelet	First-Order
Original_glszm_SizeZoneNonUniformityNormalized	Original	GLSZM
Wavelet_HHL_glszm_GrayLevelNonUniformityNormalized	Wavelet	GLSZM
LoG_-4-0-mm-3D_glcm_Idmn	LoG	GLCM
Wavelet_HHH_glszm_GrayLevelNonUniformityNormalized	Wavelet	GLSZM
LoG_-5-0-mm-3D_firstorder_Skewness	LoG	First-Order
LoG_-4-0-mm-3D_glszm_SmallAreaEmphasis	LoG	GLSZM
Gradient_firstorder_Minimum	Gradient	First-Order
Wavelet_HHL_glcm_Idn	Wavelet	GLCM

**Table 9 diagnostics-15-02546-t009:** Performance of the combined radiomics model using the top 10 selected features, without performing feature selection in Scenario 1.

Scenario 1			
Model	Classifier	AUC-PR	F1
Combined Model (Top 10 features)	Random Forest	0.860 ± 0.045	0.798 ± 0.044

**Table 10 diagnostics-15-02546-t010:** Performance of the combined radiomics model using the top 10 selected features, without performing feature selection in Scenario 2.

Scenario 2					
	Training Set			Test Set	
Model	Classifier	AUC-PR	F1	AUC-PR	F1
Combined Model (Top 10 features)	Random Forest	0.921 ± 0.059	0.868 ± 0.029	0.697	0.816

**Table 11 diagnostics-15-02546-t011:** Statistical comparison of AUC-PR values among the best-performing models across Scenario 1 using Friedman and Wilcoxon signed-rank tests. Mean differences (ΔAUC-PR) represent the average change in AUC-PR between paired models across cross-validation folds. The Friedman test was used to assess overall differences among models, while Wilcoxon tests were used for pairwise comparisons. A *p*-value < 0.05 was considered statistically significant.

Scenario 1					
Test Type	Model A	Model B	Mean ΔAUC-PR (A−B)	*p*-Value	Significance
Wilcoxon Test	Radiomics_No_ComBat	Radiomics_ADC_ratio_No_ComBat	−0.003	0.8469	No
	Radiomics_No_ComBat	Radiomics_ComBat	0.0041	0.978	No
	Radiomics_No_ComBat	Radiomics_ADC_ratio_ComBat	0.0022	0.978	No
	Radiomics_No_ComBat	Combined_model	−0.0186	0.1876	No
	Radiomics_ADC_ratio_No_ComBat	Radiomics_ComBat	0.0071	0.7197	No
	Radiomics_ADC_ratio_No_ComBat	Radiomics_ADC_ratio_ComBat	0.0051	0.5614	No
	Radiomics_ADC_ratio_No_ComBat	Combined_model	−0.0156	0.4887	No
	Radiomics_ComBat	Radiomics_ADC_ratio_ComBat	−0.0019	0.8904	No
	Radiomics_ComBat	Combined_model	−0.0226	0.2524	No
	Radiomics_ADC_ratio_ComBat	Combined_model	−0.0207	0.2769	No
Friedman Test	-	-	-	0.6325	No

**Table 12 diagnostics-15-02546-t012:** Statistical comparison of AUC-PR values among the best-performing models across Scenario 2 using Friedman, Wilcoxon signed-rank, and Nemenyi post hoc tests. Mean differences (ΔAUC-PR) represent the average change in AUC-PR between paired models across cross-validation folds. The Friedman test was used to assess overall differences among models. If significant, post hoc pairwise comparisons were performed using the Nemenyi test. Wilcoxon signed-rank tests were also applied for pairwise model comparisons. A *p*-value < 0.05 was considered statistically significant.

Scenario 2					
Test Type	Model A	Model B	Mean ΔAUC-PR (A−B)	*p*-Value	Significance
Wilcoxon Test	Radiomics_No_ComBat	Radiomics_ADC_ratio_No_ComBat	−0.0056	0.4697	No
	Radiomics_No_ComBat	Radiomics_ComBat	0.0118	0.4697	No
	Radiomics_No_ComBat	Radiomics_ADC_ratio_ComBat	0.0092	1	No
	Radiomics_No_ComBat	Combined_model	−0.0391	0.0923	No
	Radiomics_ADC_ratio_No_ComBat	Radiomics_ComBat	0.0174	0.4697	No
	Radiomics_ADC_ratio_No_ComBat	Radiomics_ADC_ratio_ComBat	0.0148	0.9097	No
	Radiomics_ADC_ratio_No_ComBat	Combined_model	−0.0335	0.0923	No
	Radiomics_ComBat	Radiomics_ADC_ratio_ComBat	−0.0026	0.7334	No
	Radiomics_ComBat	Combined_model	−0.0509	0.021	Yes
	Radiomics_ADC_ratio_ComBat	Combined_model	−0.0483	0.021	Yes
Friedman Test	-	-	-	0.0047	Yes
Nemenyi	Radiomics_No_ComBat	Combined_model	−0.0391	0.0167	Yes
	Radiomics_ComBat	Combined_model	−0.0509	0.0045	Yes
	Radiomics_ADC_ratio_ComBat	Combined_model	−0.0483	0.0523	No
	Radiomics_ADC_ratio_No_ComBat	Combined_model	−0.0335	0.1373	No

**Table 13 diagnostics-15-02546-t013:** Pairwise comparison of AUC-PR values between models on the test set using permutation testing. The ΔAUC-PR represents the difference in AUC-PR between Model A and Model B (Model A − Model B). *p*-values were derived from two-sided permutation tests with 10,000 label shuffles. A difference was considered statistically significant if *p* < 0.05. No comparisons have reached statistical significance.

Model A	Model B	ΔAUC-PR (A−B)	*p*-Value	Significance
Radiomics_No_ComBat	Radiomics_ADC_ratio_No_ComBat	−0.0051	0.8928	No
Radiomics_No_ComBat	Radiomics_ComBat	+0.0029	0.9607	No
Radiomics_No_ComBat	Radiomics_ADC_ratio_ComBat	+0.0410	0.4300	No
Radiomics_No_ComBat	ADC_ratio	+0.0486	0.4952	No
Radiomics_ADC_ratio_No_ComBat	Radiomics_ComBat	+0.0080	0.8887	No
Radiomics_ADC_ratio_No_ComBat	Radiomics_ADC_ratio_ComBat	+0.0461	0.2642	No
Radiomics_ADC_ratio_No_ComBat	ADC_ratio	+0.0537	0.4584	No
Radiomics_ComBat	Radiomics_ADC_ratio_ComBat	+0.0381	0.4976	No
Radiomics_ComBat	ADC_ratio	+0.0457	0.5329	No
Radiomics_ADC_ratio_ComBat	ADC_ratio	+0.0076	0.9177	No

## Data Availability

The local data underlying this article cannot be shared due to the privacy concerns of individuals participating in the study. The data will be shared upon reasonable request to the corresponding author. The PROSTATEx dataset used in this study is publicly available at: https://www.cancerimagingarchive.net/collection/prostatex/ (accessed on 5 October 2025).

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
