# Peer review of "Beyond Radiomics Alone: Enhancing Prostate Cancer Classification with ADC Ratio in a Multicenter Benchmarking Study"

_diagnostics, 2025, doi:10.3390/diagnostics15192546_

Round 1

Reviewer 1 Report

Comments and Suggestions for Authors

This study aims to improve the interpretation and generalizability of prostate cancer (csPCa) classification performance by integrating the ADC ratio (lesion/normal) into radiomic features derived from multiparametric MRI. Limitations frequently encountered in the literature (e.g., lack of external validation, limited modeling combinations) are systematically addressed, and the study provides a robust framework in this regard. While the article is well-organized, some aspects of the study require revision for both scientific validity and methodological soundness.
- The abstract section specifies how the ADC ratio is calculated (“lesion-to-normal”), but its definition (e.g., ROI placement) should be more clearly stated.
- AUC-PR and F1 scores are provided for performance evaluation in the experimental results, but it is essential to also include more common comparison metrics such as AUC-ROC.
- It is mentioned that ComBat harmonization did not improve discriminatory power in external validation, but sufficient information is not provided to explain why this finding is important.
- Although the introduction mentions clinical implications, the clinical significance of the ADC ratio (e.g., its relationship with necrosis and diffusion limitation) should have been explained in more depth.
- While the approach of limiting the number of models in radiomics studies is acceptable, it would have been beneficial to cite some existing benchmark studies (especially those using datasets such as TCIA and PROSTATEx).
- The introduction is quite methodologically focused. A more balanced framework could be established to appeal to the clinical reader, and the contributions of the proposed approach to the literature should be itemized in the article. - Among feature selection methods, techniques such as LASSO, RFE, and Kruskal-Wallis have been preferred, but how these methods differ from each other (e.g., filter vs. wrapper) should be more clearly explained.
- When multiple testing and model comparisons are performed, the significance of statistical differences is assessed using P-values. Furthermore, is it recommended to interpret the significance of the results using analysis of variance?
- Given the risk of overfitting, how model complexity is controlled (e.g., validation strategy) should be explained in detail.
- The finding that ComBat harmonization only improves calibration but does not increase discriminative power is stated, but the interpretation of this difference and the discussion of its potential causes are lacking.
- The results obtained, particularly whether ensemble models outperform individual classifiers, are not discussed in detail.
- While the F1 scores appear good, the presence of class imbalance is not specified. The interpretation of these metrics depends on the class distribution.
- The basis for feature selection is unclear. The most frequently selected radiomic features (GLCM, GLSZM) have not been provided with a biological correlate or clinical interpretation.

Comments on the Quality of English Language

Authors are encouraged to thoroughly revise the language of their articles.
- Some sentences in the abstract and introduction contain excessive technical jargon; accessibility for clinical readers should be improved.
- Some definitions (e.g., “Boosted GLM”) should be clarified or additional explanations provided.

Author Response

  • Comment 1: The abstract section specifies how the ADC ratio is calculated (“lesion-to-normal”), but its definition (e.g., ROI placement) should be more clearly stated.

Response 1: The ADC ratio was defined as the mean ADC of the lesion divided by that of contralateral normal tissue, by placing two identical ROIs in each side, enabling scanner-level normalization (line 28-30).

  • Comment 2: AUC-PR and F1 scores are provided for performance evaluation in the experimental results, but it is essential to also include more common comparison metrics such as AUC-ROC.

Response  2: Thank you for your comment. We agree and have now included AUC-ROC values for all models (Tables 2.1, 2.2, 3.1, 3.2, 4.1, 4.2) in the Results section text. We chose AUC-PR as the primary metric because csPCa prediction is a class-imbalanced problem where the positive class (csPCa) is of primary clinical relevance. AUC-PR is recommended in such cases as it reflects the trade-off between precision and recall for the positive class, while AUC-ROC may present overly optimistic estimates by including the true negative rate in its calculation. Nevertheless, reporting both metrics provides a comprehensive view of model performance. (line 267-271)

  • Comment 3: It is mentioned that ComBat harmonization did not improve discriminatory power in external validation, but sufficient information is not provided to explain why this finding is important.

Response 3: We modified this section of the text as follows, in order to include sufficient information:
“While ComBat harmonization improved model calibration (e.g., higher F1-scores), it did not consistently enhance external AUC-PR and occasionally reduced discriminative performance. This finding is noteworthy because ComBat is one of the most widely used statistical harmonization techniques in radiomics and is often assumed to improve cross-site generalization [48]. The most likely explanation is that ComBat reduces inter-scanner variability by aligning the means and variances of radiomic features across scanners, but this adjustment does not necessarily change the relative ordering of patient-level predictions, which is what drives AUC-PR. As a result, the model’s ability to rank csPCa versus cinsPCa cases remains similar, leaving discrimination unchanged. However, by stabilizing the feature distributions and reducing site-specific bias, ComBat makes the predicted probabilities more consistent across scanners, thereby improving probability calibration and threshold-dependent metrics such as F1-score. In some cases, this linear correction may even suppress weak biological signals if scanner effects and disease effects are partially collinear, leading to slight reductions in discriminative power on external data. This highlights that linear harmonization may be insufficient to address nonlinear scanner effects—particularly in external validation scenarios [59]. (line 549-564)

  • Comment 4: Although the introduction mentions clinical implications, the clinical significance of the ADC ratio (e.g., its relationship with necrosis and diffusion limitation) should have been explained in more depth.

Response 4: Thank you, you are right. We have now included an explanation as follows:
Lower ADC values indicate restricted diffusion from increased cellularity, whereas necrosis or atrophy may elevate ADC due to increased free water [8].The ADC ratio accentuates these contrasts by normalizing lesion ADC to patient-specific background tissue, thereby emphasizing biologically relevant diffusion differences.  (line 62-65). 

  • Comment 5: While the approach of limiting the number of models in radiomics studies is acceptable, it would have been beneficial to cite some existing benchmark studies (especially those using datasets such as TCIA and PROSTATEx).

Response 5: We appreciate this comment. We have cited the most recent benchmark study by Mylona et al. (2024), which systematically compared feature selection methods and machine-learning classifiers for prostate cancer radiomics using multicenter data. We have now also added two additional studies specifically using the PROSTATEx dataset: one evaluating workflow reproducibility and external performance [citation 37], and another investigating feature repeatability under segmentation variability [citation 38]. Together, these additions strengthen the literature review and provide a more comprehensive comparison framework for our study. We now believe that the Introduction situates our work more clearly within the context of existing benchmarks.

  • Comment 6: The introduction is quite methodologically focused. A more balanced framework could be established to appeal to the clinical reader, and the contributions of the proposed approach to literature should be itemized in the article.

Response 6: We agree with the reviewer that a more clinically balanced introduction improves readability and relevance. Accordingly, we revised the Introduction to first emphasize the clinical context, including the global burden of prostate cancer, the role of mpMRI and biopsy-derived Gleason scoring, and the limitations of PI-RADS in differentiating malignant from benign lesions. Methodological details, such as radiomics workflow, ComBat harmonization, and feature selection strategies, were streamlined into concise, accessible language to avoid overloading the clinical reader. Finally, we added a dedicated paragraph explicitly itemizing our study’s key contributions: (i) systematic benchmarking of nine feature selection methods and multiple machine-learning classifiers, (ii) evaluation of the additive value of the lesion-to-normal ADC ratio, (iii) assessment of the effect of ComBat harmonization and feature count on model performance, and (iv) robust internal and external validation on the PROSTATEx dataset. 
We now believe that these revisions provide clearer clinical motivation, highlight the novelty of our work, and ensure the introduction is accessible to both clinical and technical readers.

  • Comment 7: Among feature selection methods, techniques such as LASSO, RFE, and Kruskal-Wallis have been preferred, but how these methods differ from each other (e.g., filter vs. wrapper) should be more clearly explained.

Response 7: Thank you for this helpful comment. We have revised Section 2.3 (Feature Selection, line 227-232) to clarify the methodological differences among the applied techniques, specifying which are filter-based, wrapper-based, and embedded methods. In the Supplementary Table S2, a detailed description of each method was included.

  • Comment 8: When multiple testing and model comparisons are performed, the significance of statistical differences is assessed using P-values. Furthermore, is it recommended to interpret the significance of the results using analysis of variance?

Response 8: We appreciate this suggestion. We have clarified in the Statistical Analysis section that we used the Friedman test, which is the non-parametric analogue of repeated-measures ANOVA, to assess overall performance differences across models on the same cross-validation folds. When the Friedman test was significant, Nemenyi post-hoc tests were used to identify pairwise differences, and Wilcoxon signed-rank tests were applied for targeted comparisons. This provides a robust ANOVA-like interpretation of model performance while avoiding the assumptions of parametric ANOVA.
To better illustrate your suggestion the following was included:

Performance comparisons were conducted across validation results to identify the best-performing model configurations. To compare the performance of best-performing models across training scenarios, the Friedman test was used to assess statistically significant differences in AUC-PR values across models evaluated on the same cross-validation folds. The Friedman test is the non-parametric analogue of repeated-measures ANOVA and is recommended for comparing multiple models across cross-validation folds when normality cannot be assumed. (line 285-291)

  • Comment 9: Given the risk of overfitting, how model complexity is controlled (e.g., validation strategy) should be explained in detail.

Response 9: We explained further and in more detail the complexity as follows:
We have revised Section 2.4 to explicitly state that model complexity was controlled using a repeated nested cross-validation design, where the inner loop was used for hyperparameter tuning and feature selection, and the outer loop provided an unbiased performance estimate. This approach prevents information leakage and is recommended for high-dimensional radiomics data. Furthermore, we performed external validation on the independent PROSTATEx dataset to evaluate model generalization and detect potential overfitting.

We now included the following:

To control model complexity and reduce overfitting risk, a repeated nested cross-validation (CV) design was employed. The inner CV loop was used for hyperparameter tuning and feature selection, while the outer CV loop provided an unbiased estimate of model performance, ensuring strict separation of training and validation data. This approach is widely recommended for high-dimensional radiomics studies as it minimizes optimistic bias and prevents feature-selection leakage. (line 252-257).

  • Comment 10: The finding that ComBat harmonization only improves calibration but does not increase discriminative power is stated, but the interpretation of this difference and the discussion of its potential causes are lacking.

Response 10: We thank the reviewer for this valuable observation. We have expanded the Discussion to provide a detailed interpretation of this finding. Specifically, we explain that ComBat harmonization adjusts feature distributions by aligning their means and variances across scanners, which reduces site-specific bias and improves probability calibration (reflected in higher F1-scores) but does not change the relative ordering of patient-level predictions, leaving discriminative metrics such as AUC-PR largely unaffected. In some cases, ComBat may even attenuate subtle biological signals if scanner effects and disease effects are partially collinear, leading to slight decreases in discrimination. This new discussion appears in the revised manuscript (line 549-566).

  • Comment 11: The results obtained, particularly whether ensemble models outperform individual classifiers, are not discussed in detail.

Response 11: Thank you for pointing this out. We have expanded the Discussion to include an explicit interpretation of the ensemble results, noting that voting classifiers did not consistently outperform the best single classifiers. We also discuss possible reasons for this observation, including redundancy among base models and the near-optimal performance of the top single classifier (line 586-593).
Added: Although we explored ensemble learning through hard and soft voting classifiers, these ensembles did not consistently outperform the best single classifier across feature selection methods or validation scenarios. This outcome likely reflects the fact that the top-performing individual models already captured most of the available discrimina-tive signal, leaving limited opportunity for performance gains through aggregation. Prior radiomics studies have reported similar findings, suggesting that ensemble bene-fits are modest when base models are highly correlated or when one classifier is near optimal [63]. 

  • Comment 12: While the F1 scores appear good, the presence of class imbalance is not specified. The interpretation of these metrics depends on the class distribution.

Response 12: Thank you for this valuable observation. We have now explicitly reported the class distribution in the Patient Cohort section (line 141-142), specifying that the final dataset comprised 73.3% csPCa and 26.7% cinsPCa cases, indicating moderate class imbalance. Furthermore, we have expanded the Discussion (line 543-548) to emphasize that F1-scores should be interpreted in light of this imbalance and to highlight that AUC-PR was chosen as the primary performance metric because it better reflects the precision–recall trade-off under imbalanced conditions.

  • Comment 13: The basis for feature selection is unclear. The most frequently selected radiomic features (GLCM, GLSZM) have not been provided with a biological correlate or clinical interpretation.

Response 13: Thank you for this helpful comment. We have revised the Discussion (line 602-615) to explicitly reference Table 5 and to provide a biological interpretation of the most frequently selected features. We explain that GLCM features such as Inverse Difference Moment Normalized (IDMN) and Inverse Difference Normalized (IDN) capture gray-level homogeneity, with lower values indicating increased tissue heterogeneity, while GLSZM features such as Size-Zone Non-Uniformity Normalized (SZNN) reflect the variability of homogeneous zone sizes, a marker of architectural disorganization. These textural heterogeneity metrics have been linked to tumor aggressiveness and higher Gleason patterns, supporting their clinical relevance as imaging biomarkers. 

  • Comment 14: Some sentences in the abstract and introduction contain excessive technical jargon; accessibility for clinical readers should be improved.

Response 14: We thank the reviewer for this important observation. We have revised both the Abstract and Introduction to simplify technical language and improve accessibility for clinical readers. In the Abstract, we replaced complex methodological phrases with more concise wording and emphasized the clinical importance of accurate csPCa detection and model generalizability. In the Introduction, we shortened and clarified descriptions of radiomics workflow, harmonization, and feature selection, while maintaining methodological rigor. These changes improve readability and ensure that the clinical significance and novelty of our work are apparent to a broad readership.

  • Comment 15: Some definitions (e.g., “Boosted GLM”) should be clarified or additional explanations provided.

Response 15: We thank the reviewer for this helpful comment. In the Abstract, we have clarified this term by adding the explanation “a generalized linear model trained with boosting” (line 33) after its first mention, ensuring that readers can immediately understand the model being referred to.

Reviewer 2 Report

Comments and Suggestions for Authors

Dear Author,

In this study, the researchers used ADC maps and radiomics features for csPCa classification. Below, you can find the list of comments.
1- The article is very long and should be summarized as much as possible.
2- In this study, you have used MRI images, but the imaging parameters such as FOV, matrix size, TR, TE, imaging type, etc., have not been mentioned. It is suggested that the imaging details be added to the study.
3- The details of drawing the ROI are not included in the study and should be added. By whom were these regions identified? Was only the entire tumor used or a part of it, such as the edema areas?
4- Why was a b-value(b0)=0 used for ADC map reconstruction? Were higher values not used to reduce T2 shine-through artifacts?
5- Was fine-tuning used for the classifiers? If used, the optimal parameters and details should be added to the study.
6- The selected features  by the feature selection in each configuration should be added to the study.
7- In the introduced dataset, a very large number of samples lack biopsy information.

Best regards.

Author Response

  • Comment 1: The article is very long and should be summarized as much as possible. 

Response 1: We thank the reviewer for this comment. We carefully reviewed the manuscript and shortened the Introduction to make it more concise while ensuring that the necessary clinical and methodological context is preserved. Given the complexity of the study—covering multiple feature selection methods, classifiers, harmonization strategies, and two validation scenarios—some level of detail was necessary to maintain clarity and reproducibility. We believe the revised version strikes a better balance between completeness and brevity.

  • Comment 2: In this study, you have used MRI images, but the imaging parameters such as FOV, matrix size, TR, TE, imaging type, etc., have not been mentioned. It is suggested that the imaging details be added to the study.

Response 2: We thank the reviewer for this helpful suggestion. We have now added a detailed description of the MRI acquisition parameters, including scanner vendor, model, field strength, diffusion b-values, TR, TE, acquisition matrix, and slice thickness. This information has been summarized in a new table (Table 1) placed in the Patient Cohort section. These additions provide a clear and reproducible description of the imaging protocol used in our study.

  • Comment 3: The details of drawing the ROI are not included in the study and should be added. By whom were these regions identified? Was only the entire tumor used or a part of it, such as the edema areas?

Response 3: Thank you for this valuable comment. We have revised Section 2.1 (Patient Cohort) to include a more detailed description of ROI delineation. Lesions were retrospectively segmented on ADC maps by a board-certified radiologist with more than 20 years of experience using LIFEx software, incorporating T2w and DWI images for anatomical guidance and correlating with histopathology. We now clarify that segmentation was performed on the entire visible lesion, excluding surrounding benign tissue, peritumoral edema, and post-biopsy hemorrhage, to ensure that the ROI accurately represented tumor tissue without confounding signal from adjacent structures. (line 125-131)

  • Comment 4: Why was a b-value(b0)=0 used for ADC map reconstruction? Were higher values not used to reduce T2 shine-through artifacts?

Response 4: Thank you for this question. In both the local multicenter dataset and the PROSTATEx public dataset, ADC maps were reconstructed using a low b-value (0 or 50 s/mm²) together with a higher b-value of 800 s/mm², in accordance with PI-RADS v2-compliant prostate MRI protocols. This approach reflects standard clinical practice and ensures consistency between datasets. While higher minimum b-values can reduce T2 shine-through, inclusion of b0/b50 provides higher SNR and supports robust mono-exponential ADC fitting.

  • Comment 5: Was fine-tuning used for the classifiers? If used, the optimal parameters and details should be added to the study.

Response 5: Thank you for this helpful comment. We confirm that hyperparameter tuning was performed for all classifiers using grid search within the inner cross-validation loop, optimizing for mean AUC-PR. We have added Supplementary Tables S6 and S7 summarizing the final tuned hyperparameters for the best-performing models in Scenarios 1 and 2. In the Results section (line 336-339) we now refer readers to these supplementary tables for full details.

  • Comment 6: The selected features  by the feature selection in each configuration should be added to the study.

Response 6: Thank you for this comment. Given the large number of configurations evaluated (nine feature selection methods, multiple classifiers, harmonization schemes, and repeated cross-validation folds), it is not feasible to report the selected features for each individual configuration. Instead, we have rephrased Section 3.8 (line 452-463) to highlight consistent patterns and present in Table 5 the ten most frequently selected features across all models, representing the most robust and stable radiomic biomarkers identified in our study.

  • Comment 7: In the introduced dataset, a very large number of samples lack biopsy information.

Response 7: We appreciate this comment and clarify that all lesions included in our study were biopsy-confirmed. In the local dataset, patients underwent both TRUS-guided systematic biopsy and MRI-targeted biopsy, while in the PROSTATEx dataset, histopathological evaluation was performed by an experienced uropathologist as described in the original dataset publication. Lesions without biopsy confirmation or with incomplete records were excluded. 

Round 2

Reviewer 1 Report

Comments and Suggestions for Authors

It can be said that the authors fully fulfill the desired corrections and give satisfactory answers to the comments. Paper has been improved significantly and I think it does not need to process more.

Author Response

We sincerely thank you for your constructive feedback and positive evaluation of our revised manuscript. We greatly appreciate your acknowledgment that the requested corrections were fully addressed and that the paper has been significantly improved.

Reviewer 2 Report

Comments and Suggestions for Authors

Dear Author,

thanks for providing revised manuscript. The authors answered to all my comments and now I do not have any further comments.

Best Regards

Author Response

We are very grateful for your careful review and for confirming that our revisions satisfactorily addressed all of your comments. Your feedback was invaluable in refining and improving the manuscript.

Best Regards